# How did we protect children against COVID-19 in Iran? Prevalence of COVID-19 and vaccination in the socio-economic context of COVID-19 epidemic

**Meroe Vameghi[1], Mohammad Saatchi[2,3]\*, Giti Bahrami[4], Farin Soleimani[5], Marzieh Takaffoli[1]**

1 Social Welfare Management Research Center, Social Health Research Institute, University of Social Welfare and Rehabilitation Sciences, Tehran, Iran, 2 Department of Biostatistics and Epidemiology, University of Social Welfare and Rehabilitation Science, Tehran, Iran, 3 Health in Emergency and Disaster Research Center, Social Health Research Institute, University of Social Welfare and Rehabilitation Sciences, Tehran, Iran, 4 Social Determinants of Health Research Center, Alborz University of Medical Sciences, Karaj, Iran, 5 Paediatric Neurorehabilitation Research Centre, University of Social Welfare and Rehabilitation Sciences, Tehran, Iran

\* m.saatchi65@gmail.com

**Data Availability Statement:** All relevant data are within the manuscript.

## Abstract

### Introduction

The COVID-19 pandemic posed significant risks to children worldwide. This study aimed to assess the COVID-19 protection status of children and explored the relationship between household socio-economic status and COVID-19 morbidity and preventive measures, including vaccination and mask-wearing, in two cities in Iran.

### Method

A population-based cross-sectional study was conducted from July to October 2022 among 7 to 18-year-old children and their families in Tehran and Karaj. A total of 3,022 samples were selected using stratified multistage cluster sampling. Data were collected through interviews with children and adults, using questionnaires and was analyzed with Stata software version 14.

### Results

The analysis focused on 2,878 children with a median age of 12. Over half (54%) reported that the pandemic negatively affected their family's financial status, with 45% describing its impact on children's needs as negative or very negative. Just under 50% of respondents consistently wore masks during the study period, and around 54% had received at least one dose of the COVID-19 vaccine. Reasons for not getting vaccinated included concerns about side effects, ineligibility for the target age group, and overcrowding at vaccination sites. The odds of not getting vaccinated were significantly lower for children aged 15–18, with boys more likely to refuse vaccination than girls.

**Funding:** This study was sponsored by the University of Social Welfare and Rehabilitation Science [contract number: 1401/801/A/11799 ] (https://uswr.ac.ir/) recivied by Meroe Vameghi and Alborz University of Medical Sciences, Karaj, Iran [contract number: 1401/60/D/1495] (https://abzums.ac.ir/) received by Giti Bahrami. Moreover, no additional external funding was received for this study.

**Competing interests:** The authors have declared that no competing interests exist.

## Conclusion

The financial impact of the pandemic in Iran affected families' ability to meet their children's needs. Moreover, low vaccination acceptance rates increased children's vulnerability to health problems and contributed to COVID-19 infections. Efforts should be made to increase vaccination acceptance, particularly among immigrant populations.

## Introduction

COVID-19 was one of the world's most significant health challenges, with major effects on various aspects of communities and the health of adults and children [1, 2]. Out of the 4.4 million deaths from COVID-19 till January 2023 reported in the MPIDR (Max Planck Institute for Demographic Research) Coverage database from 118 countries, 0.4% (more than 17,400) deaths occurred among children and adolescents under the age of 20, with 53% being among adolescents aged 10 to 19 and 47% of children aged 0 to 9 [3]. In addition, infected but asymptomatic children are important in transmitting the disease to adults [4]. These data showed that, unlike the initial impressions at the start of the pandemic, children are at risk of transmission, infection, and death.

As the UN [5] announced, the COVID-19 pandemic affects children in three ways: (a) direct effects on the health of children and their families, (b) immediate changes resulting from restrictive measures, such as the closure of group activities to prevent and treat disease transmission, and (c) the long-term impact of the crisis on the achievement of the Sustainable Development Goals. The indirect effects of the pandemic on children's survival and health are due to the changes to the health system and the suspension of preventive care services [6], reduced access to child and pregnant mothers' health care, increased child malnutrition, lack of access to drinking water, health, and sanitation, reduced access to HIV prevention and treatment services, decreased sexual health, increased adolescent fertility and decreased mental health of children and adolescents [3, 7, 8].

Research shows that these outcomes are different in different societies and communities. Children living in families with low socio-economic status experienced the most harm in the coronavirus pandemic. However, other factors related to individual and family characteristics were influential [9]. According to a study, approximately 11% of COVID-19 cases among people younger than 20 were observed in low- and middle-income countries compared to 7% in high-income countries. According to the study, the prevalence varies widely in different countries, ranging from 23% of COVID-19 cases in Paraguay to 0.82% in Spain [10].

On the other hand, vaccination and wearing masks are major strategies to prevent the spread of the COVID-19 virus. However, as the World Health Organization emphasizes, vaccine hesitancy threatens global health and is a severe challenge in the fight against COVID-19 [11]. Previous studies showed that parents' intentions to vaccinate children generally range from 60% to 70% due to parental skepticism, concerns about the vaccine's safety, and disinformation or misinformation related to vaccination [12–14].

In Iran, the COVID-19 pandemic was recognized in February 2020, and by the end of March 2022, Iran had experienced eight waves of COVID-19. Vaccinations for children aged 12 to 18 and 5 to 12 against COVID-19 began in September 2021 and February 2022, respectively [15]. As in the rest of the world, the pandemic, directly and indirectly, affected the children of Iran through all the measures and protocols related to disease prevention and control, especially the closure of schools.

Approximately 14 million students in Iran were at risk of COVID-19 [16]. Previous studies conducted on children in Iran have mainly addressed the clinical signs and symptoms among children [17–19]. Thus, there is a need to investigate the non-clinical factors related to the morbidity and prevention of COVID-19, given the limited evidence about the prevalence of COVID-19 in children in Iran and the necessity of timely interventions to prevent and mitigate the consequences of the pandemic on children are needed. We conducted this study to investigate the relationship between the socio-economic status of families and their response to COVID-19, focusing on childhood vaccination in the general population of children in Tehran and Karaj.

## Materials and methods

### Study setting

The study was conducted in Tehran and Karaj from July to October 2022. Tehran is the capital of Iran, with a population of 9 million and 22 districts. Karaj is the center of Alborz province and the largest immigrant hub of Iran after Tehran, with a population of about 1.6 million and 12 districts. Both cities are located in the center of Iran and are known as cities with high cultural and social diversity of inhabitants [20].

### Study population and sampling method

The present study is a descriptive-analytical cross-sectional one conducted to assess the health and social consequences of the COVID-19 pandemic in children. The target population was 7-18-year-old children living in Tehran and Karaj cities and their families. The study samples were selected using multistage cluster stratified sampling, which included a child aged 7 to 18 years plus one adult member of selected families (only for the age group of 7 to 14 years) who lived in the same home with other family members during the COVID-19 pandemic. In Tehran and Karaj, the largest urban division unit is the district, and each district comprises several areas, and each area is composed of several neighborhoods. In our study, each of the 22 regions of Tehran and 12 in Karaj city was considered as one stratum, and from each stratum, one area, and from each area, a neighborhood was selected as clusters. In each cluster, the household was the study unit. The number of selected households in each cluster was determined according to previous population-based studies conducted in Tehran [21], with 20 households per cluster. The sample size of each district in Tehran was calculated based on the proportion of the children in each. Since no information was available about the number of children in each district in Karaj, the sample size of all districts was assumed to be equal. The required sample size was calculated based on the ratio estimation formula. Considering the prevalence of 9% for estimation of the occurrence of Coronavirus in children [22] and absolute error value of 0.018, the probability of loss to flow up was 25%, and considering the design effect of 1.5, the minimum sample size was 2062 children in Tehran and 960 in Karaj.

### Data collection and measurement

Data were collected through questionnaires at the participants' residences. The questionnaire takers had already been trained in a two-hour workshop about study objectives, sampling methodology, completing a questionnaire, and complying with research ethics. The questionnaire takers referred to each neighborhood where a unit in a building was selected. They introduced themselves and asked the residents about the number and age of the children in the family. Then, they would choose a child randomly to complete the questionnaire. If the selected child was between 7 and 14 years old, the mother or father or one of the informed

adults of the house was interviewed, and the questionnaires were filled out. The informants were interviewed if the child was between 15 and 18 years old. Data collection tools were two researcher-made questionnaires, including information about the family status and information for children 7–14 and 15- -18 years old. Experts in the field were asked to review the questionnaire for content validity. The mean CVR and CVI were 0.84 and 0.79, respectively, indicating acceptable validity of the questionnaire.

## Variables

Study variables were measured in three parts. The first part dealt with demographic characteristics of the child, e.g., age, gender, and nationality; the second part covered the socio-economic status of the family, e.g., family size, education and employment status of parents, house ownership, family income sources in addition to that of the guardian, family financial status, pandemic impact on the family economy, and the family's financial ability to meet the child's needs based on self-assessment, COVID-19, and having health insurance. The third part included variables related to the Protective behaviors of children and families against COVID-19 infection, e.g., use of masks, getting vaccinated, willingness to get vaccinated in the future, COVID-19's impact on the child's ability to earn income, the COVID-19 infection and how it was diagnosed, hospitalization due to COVID-19, and access to health services. We look at these variables in the socio-economic context of families during the COVID-19 pandemic in Iran.

## Statistical analysis

The data were analyzed descriptively and analytically, following the study's objectives. Mean and standard deviation were used to report continuous variables; frequency and percentages were used for the categorized variables. The variables for Tehran and Karaj were analyzed separately and presented as a whole. COVID-19 morbidity and lack of vaccination were the two primary outcomes of the analytical part of the study, which was used to determine the factors associated with them. First, univariate independent variables were used for the logistic regression model, and the values of unadjusted Odds Ratio and 95% confidence interval were calculated for each independent variable. The variables with p-values less than 0.2 were selected, and multivariable analysis was conducted using backward stepwise logistic regression. Data were analyzed using Stata software version 14 with a significance level of 0.05.

## Ethical issues

This study was approved by the Ethics Committee of the University of Social Welfare and Rehabilitation Sciences (IR.USWR.REC.1401.001). Participants were assured that their personal information and answers would remain confidential.

To complete questionnaires for the children aged 7 to 14, we obtained informed written consent from the interviewed parent or adult. To complete questionnaires for the children who were 15–18 years old, we obtained informed written consent to interview the child from both the child and the parent or adult who was at home. For this age group, the interview could be conducted in the presence of the adult or alone with the child due to adult permission. The study's objectives were clearly explained to adults and children.

## Results

The participants (n = 2878) were 7 to 18 years old. The sample included 2057 children aged 7 to 14 (71.5%) and 821 youngsters aged 15 to 18 (28.5%) from 22 districts of Tehran and 12 in

Karaj. The median age of the study participants was 12 years, with an equal number of boys and girls. Of the sample, 8.4% of the informants were Afghan, and the rest were Iranian. Most respondents were mothers in the age group 7–14 years old, approximately 6% of whom were illiterate. Table 1 shows the baseline characteristics of children in the age groups 7–14 and 15–18, separately for Tehran and Karaj.

Considering the critical role of financial status in physical health and adherence to health care, Table 2 summarizes the financial status indicators of Tehran and Karaj's participating households in the study. According to the results, less than 3% of the heads of households described themselves as "always unemployed"; approximately 68.5% described themselves as "always employed." Child labor was cited as a source of income by 4.3% of households. The number of households using child labor as a source of income in Karaj was twice that of Tehran. 22.5% of participants reported their financial situation as poor and very poor, and about 54% of the participants in Tehran and Karaj evaluated the impact of COVID-19 on their families' economies as negative and very negative. About 44% of participants noted that their family's financial ability to meet their child's needs is negative or very negative.

## Prevention of COVID-19

The prevalence of mask-wearing from July to October 2022 was 46.7% (95% CI: 44.9 to 48.5), which was 49% in Tehran (95% CI: 46.8 to 51.3) and 42.2% in Karaj (95% CI: 39.1% to 45.3). Mask-wearing in the age group of 7–14 in Tehran was significantly more compared to those living in Karaj (P-value≤0.001), but there was no significant difference (P-value = 0.652) in the age group of 15–18 concerning their place of residence (Fig 1).

The most common reasons for children not wearing masks were believing that COVID-19 was decreasing or ending (38.9%) and intolerance (36.7%). Other reasons included thinking that wearing a mask was not effective in preventing Coronavirus (14.7%), not being convinced about the outbreak of Coronavirus (3.8%), high cost of providing masks (3.1%), family preference for using traditional medicine (1.3%), and other causes (1.5%). Fig 2 shows how these reasons varied by age group and city of residence.

By the time the study was carried out, 54% of participants had received at least one dose of the COVID-19 vaccine. This proportion in Tehran and Karaj was 61% (95% CI: 58.9 to 63.3%) and 41% (95% CI: 37.9 to 44.1%), respectively (P-Value = 0.001). Moreover, 42% (95% CI: 40.0 to 44.7%) of the respondents were in the age group of 7–14, and 83.2% of the children (95% CI: 80.0 to 85.6%) were in the age group of 15–18 years old had been vaccinated, respectively (P-Value = 0.001). In the age group of 7–14 years old who received the vaccine, 61% received two doses, compared to 77% in 15 -18-year-old children. In the sample (n = 2878), 36.4% of the participants received two vaccine doses. In the age group of 7–14, 25.7%, and 15- -18, 63.4% of the children received two vaccine doses. Approximately 82% of the vaccinated children received at least one dose of China's Sinopharm vaccine and 4.1% of the children received at least one dose of Iran 's Barekat vaccine. AstraZeneca, Sputnik, Pfizer, and Pastocovac (13.9%) were among the other vaccines used sparsely. In general, concerns about the complications of the vaccine, not being included in the age group, and the over-crowdedness of the injection sites were the three main reasons cited for non-vaccination at the time of the study (Table 3). The willingness to get the COVID-19 vaccine in the participants who have not been vaccinated until the time of conducting the survey, by city and origin, is shown in Fig 3. The report of definitely unwilling among Iranians living in Tehran and Karaj was 65% and 29%, respectively, and among Afghans, it was 41% and 24%, respectively.

Table 4 shows the factors affecting the non-vaccination against COVID-19 in children using univariate and multivariable logistic regression. In the age group of 15–18 years old,

**Table 1. Frequency distribution of baseline characteristics in children aged 7–14 and 15–18 in total and separately for Tehran and Karaj.**

| Variable | Total | | Tehran | | Karaj | |
|---|---|---|---|---|---|---|
| | N | % | N | % | N | % |
| **Child age group** | | | | | | |
| 7–14 | 2057 | 71.5 | 1356 | 71.8 | 701 | 71 |
| 15–18 | 821 | 28.5 | 533 | 28.2 | 288 | 29 |
| **Child sex** | | | | | | |
| Female | 1438 | 50.4 | 934 | 50.1 | 504 | 51 |
| Male | 1415 | 49.6 | 931 | 49.9 | 484 | 49 |
| **Nationality of origin** | | | | | | |
| Iranian | 2607 | 91.6 | 1757 | 94.0 | 850 | 87 |
| Afghan | 238 | 8.4 | 111 | 6.0 | 127 | 13 |
| **Respondent** | | | | | | |
| Mother | 1332 | 65.8 | 892 | 66.6 | 440 | 64.2 |
| Father | 303 | 15.0 | 222 | 16.6 | 81 | 11.8 |
| Others | 389 | 19.2 | 225 | 16.8 | 164 | 24.0 |
| **Father's education** | | | | | | |
| Illiterate | 171 | 6.0 | 63 | 3.3 | 108 | 10.9 |
| Primary and secondary | 620 | 21.6 | 322 | 17.1 | 298 | 30.2 |
| High school and Diploma | 1092 | 38.0 | 740 | 39.3 | 352 | 35.7 |
| Academic | 985 | 34.4 | 756 | 40.2 | 229 | 23.2 |
| **Mother's education** | | | | | | |
| Illiterate | 178 | 6.2 | 78 | 4.1 | 100 | 10.1 |
| Primary and secondary | 556 | 19.3 | 253 | 13.4 | 303 | 30.7 |
| High school and Diploma | 1101 | 38.3 | 719 | 38.1 | 382 | 38.7 |
| Academic | 1036 | 36.4 | 835 | 44.4 | 201 | 20.3 |
| **Head of Family** | | | | | | |
| Father | 2680 | 93.5 | 1779 | 95.6 | 881 | 89.4 |
| Mother | 163 | 5.7 | 77 | 4.1 | 86 | 8.7 |
| Other | 24 | 0.8 | 5 | 0.3 | 17 | 1.9 |
| **The job of the head of the family** | | | | | | |
| employee | 795 | 26.7 | 580 | 30.8 | 215 | 21.8 |
| Worker | 585 | 20.4 | 244 | 12.9 | 341 | 34.5 |
| Farmer | 14 | 0.49 | 2 | 0.1 | 12 | 1.2 |
| Retired | 154 | 5.4 | 69 | 3.7 | 85 | 8.6 |
| Self-employed | 1140 | 39.7 | 857 | 46.6 | 265 | 26.8 |
| Housewife | 31 | 1.08 | 10 | 0.5 | 21 | 2.1 |
| Other | 154 | 5.36 | 102 | 5.5 | 48 | 4.9 |
| **Homeownership status** | | | | | | |
| Owner | 1430 | 49.8 | 989 | 52.5 | 441 | 44.7 |
| Rental | 1241 | 43.2 | 794 | 42.1 | 447 | 45.3 |
| Employee housing | 37 | 1.29 | 18 | 1.0 | 19 | 1.9 |
| Other | 163 | 5.7 | 83 | 4.4 | 78 | 8.1 |
| **Child Medical Insurance** | | | | | | |
| No medical insurance | 709 | 24.7 | 408 | 21.7 | 301 | 30.5 |
| Basic | 1614 | 56.2 | 1068 | 56.7 | 546 | 55.4 |
| Basic and complementary | 545 | 19.1 | 406 | 21.6 | 139 | 14.1 |

**Table 2. Financial variables of households in Tehran and Karaj.**

| Variable | Total | | Tehran | | Karaj | |
|---|---|---|---|---|---|---|
| | N | % | N | % | N | % |
| **Family head occupational status** | | | | | | |
| Unemployed | 73 | 2.6 | 41 | 2.2 | 32 | 3.2 |
| Employed | 1960 | 68.4 | 1443 | 76.6 | 517 | 52.5 |
| Sometimes employed | 689 | 24 | 335 | 17.9 | 354 | 35.9 |
| Retired | 144 | 5.0 | 62 | 3.3 | 82 | 8.3 |
| **Having other income sources** | | | | | | |
| No other income source | 1642 | 57.4 | 1162 | 61.9 | 480 | 48.9 |
| Working of other family members except the child | 969 | 33.9 | 601 | 32.0 | 368 | 37.5 |
| The child | 123 | 4.3 | 58 | 3.1 | 65 | 6.6 |
| Relatives | 33 | 1.1 | 10 | 0.5 | 23 | 2.3 |
| Imam Khomeini Relief Foundation | 17 | 0.6 | 4 | 0.2 | 13 | 1.3 |
| State Welfare organization | 19 | 0.7 | 13 | 0.7 | 6 | 0.6 |
| Unemployment insurance | 10 | 0.4 | 5 | 0.3 | 5 | 0.5 |
| Charities | 25 | 0.9 | 16 | 0.9 | 9 | 0.9 |
| Other | 21 | 0.7 | 9 | 0.6 | 12 | 1.2 |
| **Family financial status** | | | | | | |
| Very good | 103 | 3.6 | 50 | 2.7 | 53 | 3.6 |
| Good | 642 | 22.4 | 429 | 22.8 | 213 | 22.4 |
| Medium | 1476 | 51.4 | 1002 | 53.2 | 474 | 51.4 |
| Bad | 516 | 17.9 | 304 | 16.1 | 212 | 18.0 |
| Very bad | 133 | 4.6 | 99 | 5.3 | 34 | 4.6 |
| **Covid 19 Pandemic effect on family financial status** | | | | | | |
| Very positive | 6 | 0.2 | 3 | 0.2 | 3 | 0.3 |
| positive | 54 | 1.8 | 45 | 2.4 | 9 | 0.9 |
| No effect | 1185 | 41.4 | 781 | 41.6 | 404 | 41 |
| Negative | 1039 | 36.3 | 677 | 36.1 | 362 | 36.7 |
| Very negative | 504 | 17.6 | 331 | 17.6 | 173 | 17.5 |
| I don't know | 74 | 2.5 | 39 | 2.1 | 35 | 3.5 |
| **COVID-19 Pandemic effect on meeting child needs** | | | | | | |
| Very positive | 11 | 0.4 | 7 | 0.4 | 4 | 0.4 |
| positive | 50 | 1.7 | 40 | 2.1 | 10 | 1.0 |
| No effect | 1423 | 49.6 | 967 | 51.4 | 456 | 46.3 |
| Negative | 897 | 31.3 | 608 | 32.3 | 289 | 29.4 |
| Very negative | 403 | 14.0 | 215 | 11.4 | 188 | 19.1 |
| I don't know | 82 | 2.9 | 45 | 2.4 | 37 | 3.8 |
| **Child working due to Covid 19 Pandemic** | | | | | | |
| Yes | 88 | 3.1 | 50 | 2.7 | 38 | 3.9 |
| No | 2758 | 96.9 | 1821 | 97.3 | 937 | 96.1 |

non-vaccination was about 80% lower than the age group of 7–14 years old (P-value = ≤0.001), and boys' odds ratio was 1.3 compared to girls (P-Value = 0.005). Adjusted for confounding variables, the odds ratio of non-vaccinated children living in Karaj was approximately 3.5 compared to Tehran (P-value = ≤0.001). The odds ratio of non-vaccination of Afghan children was 2.77 compared to Iranian children (P-value = ≤0.001). The mother's education and financial status were among the factors contributing to non-vaccination in children.

**Fig 1. Frequency distribution of mask-wearing in children by city of residence and age group.**

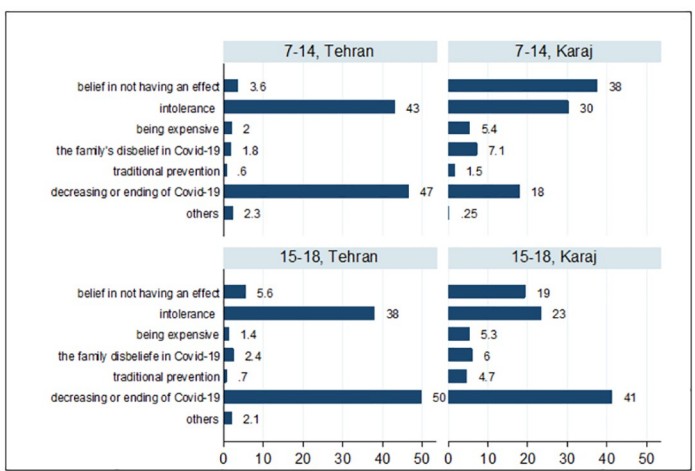

**Fig 2. Distribution of causes of children not wearing masks by age group and city of residence.**

**Table 3. The reasons for non-vaccination of children in the study.**

| Variables | N | %(CI) |
|---|---|---|
| **Vaccine hesitancy reasons** | | |
| Worry about vaccine side effects | 524 | (42.0–47.7) 44.8 |
| Worry about vaccination site crowding | 101 | (7.1–10.4) 8.6 |
| The child is not infected with Covid 19 | 37 | (2.3–4.3) 3.1 |
| Pandemic has finished | 9 | (0.4–1.4) 0.7 |
| Vaccine Ineffectiveness | 79 | (5.4–8.3) 6.7 |
| The child is too young for vaccination | 418 | (29.9–35.2) 32.6 |
| Non-availability of vaccine | 14 | (0.7–2.0) 1.1 |
| Other | 24 | (1.3–3.1) 2.0 |
| Total | 1206 | 100 |
| **Vaccine acceptance in future** | | |
| Yes | 124 | (8.6–12.0) 10.2 |
| No | 626 | (48.6–54.2) 51.4 |
| Uncertain | 464 | (35.5–41.0) 38.2 |
| **Total** | 1214 | 100 |

Our findings showed that 20.1% of the respondents caught COVID-19 from July to October 2022 (CI 95%:18.6 to 21.6%); (2.6 percent based on a PCR test, 6.4 percent with a doctor's diagnosis without a PCR test, and 11.1 percent based only on clinical symptoms and 79.9% non-infected). Approximately 17% of the age group 7–14 years old (95% CI: 15.3 to 18.5) and 28% of the age group 15–18 caught COVID-19 (95% CI: 25.1% to 31.3%); 1.39% of the respondents said they had been hospitalized (95% CI: 1.02 to 1.88%). Among the age groups of 7–14 and 15–18 years old, the prevalence of hospitalization was 0.94% (95% CI: 0. 64 to 1.5%) and 2.4% (95% CI: 1.5 to 3.7%), respectively.

A total of 1,229 (42.7%) children required health care from July to October 2022, of whom 310 children (25.5%) required COVID-19-related health services, while 114 children (9.6%) did not receive any assistance. This rate was 22.5% in Tehran and 5.2% in Karaj. Fig 4 shows the most important causes of non-referral to receive services.

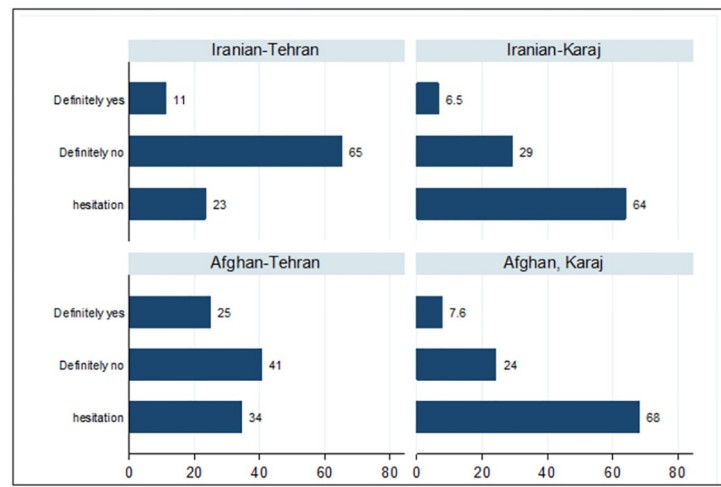

**Fig 3. The percentage of willingness to get vaccinated in Iranian and Afghan participants based on city.**

**Table 4. Factors affecting COVID-19 non-vaccination in children using univariate and multivariate logistic regression.**

| Variable | Unadjusted OR(95% CI) | P-Value | Adjusted OR | P-Value |
|---|---|---|---|---|
| **Child age group** | | ≤0.001 | | ≤0.001 |
| 7–14 | 1.00 | | 1.00 | |
| 15–18 | 0.22 (0.17, 0.27) | | 0.19(0.15, 0.24) | |
| **Child sex** | | 0.012 | | 0.005 |
| Female | 1.00 | | 1.00 | |
| Male | 1.23(1.04, 1.45) | | 1.30 (1.08, 1.56) | |
| **City** | | ≤0.001 | | ≤0.001 |
| Tehran | 1.00 | | 1.00 | |
| Karaj | 2.95(2.48, 3.50) | | 3.47 (2.83, 4.24) | |
| **Nationality of origin** | | ≤0.001 | | ≤0.001 |
| Iranian | 1.00 | | 1.00 | |
| Afghan | 2.65 (1.99, 3.35) | | 2.77 (1.78, 4.30) | |
| **Father's education** | | | | |
| Academic | 1.00 | | | |
| Illiterate | 2.80(1.97, 3.98) | ≤0.001 | - - - - | - - - - |
| Primary and secondary | 1.48 (1.19, 1.85) | ≤0.001 | - - - - | - - - - |
| High school and diploma | 1.09 (0.89, 1.33) | 0.339 | - - - - | - - - - |
| **Mother's education** | | | | |
| Academic | 1.00 | | 1.00 | |
| Illiterate | 2.17(1.55, 3.05) | ≤0.001 | 0.55 (0.32, 0.93) | 0.026 |
| Primary and secondary | 1.50 (1.19, 1.87) | ≤0.001 | 0.81 (0.61, 1.07) | 0.154 |
| High school and diploma | 1.01 (0.83, 1.23) | 0.866 | 0.79 (0.63, 0.99) | 0.042 |
| **Family size** | | ≤0.001 | | |
| 2–4 | 1.00 | | | |
| Five and more | 1.37(1.15,1.64) | | - - - - | - - - - |
| **Head of family job** | | | | |
| Self-employed | 1.00 | | | |
| employee | 0.96(0.78–1.18) | 0.719 | - - - - | - - - - |
| Worker | 2.05(1.65–2.55) | ≤0.001 | - - - - | - - - - |
| Farmer | 2.28(0.76–6.84) | 0.141 | - - - - | - - - - |
| Retired | 1.03(0.71–1.49) | 0.873 | - - - - | - - - - |
| Housewife | 1.15(0.52–2.54) | 0.728 | - - - - | - - - - |
| Other | 1.09(0.74–1.60) | 0.641 | - - - - | - - - - |
| **Head of the family occupational status** | | | | |
| Employed | 1.00 | | | |
| Unemployed | 1.27 (0.76–2.12) | 0.347 | - - - - | - - - - |
| Sometimes employed | 1.49(1.23–1.80) | ≤0.001 | - - - - | - - - - |
| Retired | 1.12 (0.78–1.61) | 0.530 | - - - - | - - - - |
| **Self-report Family financial status** | | | | |
| Medium | 1.00 | | 1.00 | |
| Good and very good | 0.72(0.58–0.88) | 0.002 | 0.72(0.57–0.91) | 0.006 |
| Bad and very bad | 1.69(1.38–2.06) | ≤0.001 | 1.43(1.13–1.81) | 0.003 |
| **Covid 19 Pandemic effect on family financial status** | | | | |
| Negative and very negative | 1.00 | | | |
| Positive and very positive | 0.38(0.19–0.75) | 0.005 | - - - - | - - - - |
| No effect | 0.88(0.74–1.04) | 0.141 | - - - - | - - - - |

*(Continued)*

**Table 4.** (Continued)

| Variable | Unadjusted OR(95% CI) | P-Value | Adjusted OR | P-Value |
|---|---|---|---|---|
| **Child working due to the pandemic** | | 0.279 | | |
| No | 1.00 | | | |
| Yes | 1.28(1.37–3.39) | | - - - - | - - - - |
| **Method of education** | | 0.021 | | |
| Face-to-face and mix of face-to-face and online | 1.00 | | | |
| Online | 0.75 (0.58–0.95) | | - - - - | - - - - |

Table 5 shows the factors affecting children's protection against COVID-19 using univariable and multiple variables logistic regression. The final model shows that the influencing factors were (a) to which age group the respondent belonged, (b) how many people lived with the child, (C) whether the head of the household was the breadwinner or not, (d) whether the head of the household was employed or not, (e) what was the family's self-assessed financial status, (f) whether the family used child labor, and (g) whether and how the family was educated about COVID-19. In the final model, the child's sex, place of residence, nationality, occupation of the household head, retirement of the household head, and the positive effect of the Coronavirus on the family economy was not significant.

## Discussion

This study is one of the few studies carried out in Iran that has examined the prevalence of COVID-19 and vaccination against COVID-19 in the general population of children using a representative sample. The focus of the study was how protection against COVID-19 and vaccine hesitancy in children was related to the socio-economic status of families. This study investigated two ways to prevent COVID-19 infection: wearing masks in public spaces and getting vaccinated. It should be noted that, at the time of the study, Iran was in the seventh wave of COVID-19, dominated by the Omicron variant, while the schools began to operate fully onsite for the first time since the pandemic.

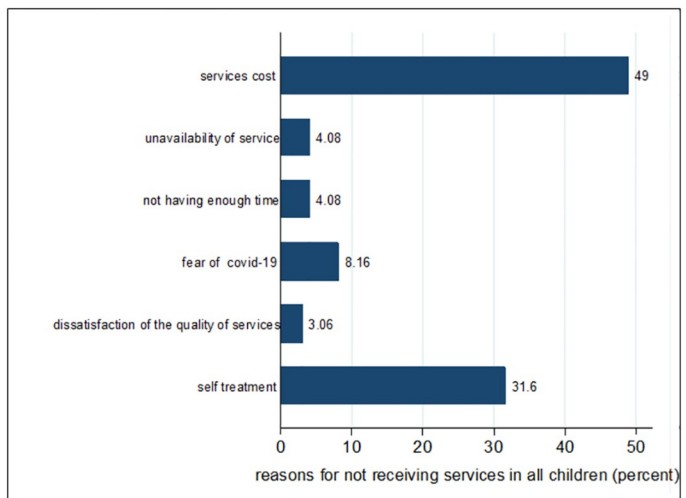

**Fig 4. The most important reasons for not receiving services in all children aged 7 to 18.**

**Table 5. Factors affecting the contraction of COVID-19 in children using univariate and multivariate logistic regression.**

| Variable | Unadjusted OR(95% CI) | P-Value | Adjusted OR | P-Value |
|---|---|---|---|---|
| **Child age group** | | 0.001 | | 0.001 |
| 7–14 | 1.00 | | 1.00 | |
| 15–18 | 1.92(1.58, 2.33) | | 1.72 (1.31, 2.06) | - - - - |
| **Child sex** | | 0.254 | | |
| Female | 1.00 | | - - - - | |
| Male | 1.11(0.92, 1.33) | | - - - - | - - - - |
| **City** | | 0.539 | | |
| Tehran | 1.00 | | - - - - | |
| Karaj | 0.94(0.77, 1.14) | | - - - - | - - - - |
| **Nationality of origin** | | 0.010 | | |
| Iranian | 1.00 | | - - - - | |
| Afghan | 0.60(0.41, 0.88) | | - - - - | - - - - |
| **Family size** | | 0.001 | | |
| 2–4 | 1.00 | | 1.00 | |
| Five and more | 0.57(0.46–0.70) | | 0.61(0.49–0.81) | - - - - |
| **Father's education** | | | | |
| Academic | 1.00 | | | |
| Illiterate | 0.55(0.31, 0.88) | 0.007 | - - - - | - - - - |
| Primary and secondary | 0.63 (1.19, 1.85) | 0.001 | - - - - | - - - - |
| High school and diploma | 0.85 (0.89, 1.33) | 0.166 | - - - - | - - - - |
| **Mother's education** | | | | |
| Academic | 1.00 | | 1.00 | |
| Illiterate | 0.51(0.31, 0.87) | 0. 007 | 0.60 (0.30, 1.23) | 0.171 |
| Primary and secondary | 0.61 (0.46, 0.82) | 0.001 | 0.62 (0.42, 0.92) | 0.019 |
| High school and diploma | 1.01 (0.80, 1.26) | 0.930 | 1.01 (0.78, 1.32) | 0.894 |
| **Family head job** | | | | |
| Self-employed | 1.00 | | | |
| Employee | 1.17(0.94–1.46) | 0.153 | - - - - | - - - - |
| Worker | 0.65(0.49–0.85) | 0.002 | - - - - | - - - - |
| Farmer | 1.08(0.81–3.30) | 0.869 | - - - - | - - - - |
| Retired | 1.77(1.21–2.57) | 0.003 | - - - - | - - - - |
| Housewife | 2.19(1.03–4.60) | 0.040 | - - - - | - - - - |
| Other | 0.75(0.47–1.19) | 0.231 | - - - - | - - - - |
| **Family head occupational status** | | | | |
| Employed | 1.00 | | | |
| Unemployed | 2.30(1.38–3.81) | 0.001 | 2.62(1.45–5.72) | 0.003 |
| Sometimes employed | 1.23(0.99–1.53) | 0.056 | 1.52(1.27–2.29) | 0.009 |
| Retired | 2.20 (1.52–3.11) | 0.001 | 2.13 (0.89–6.92) | 0.001 |
| **Self-report financial status** | | | | |
| Good and very good | 1.00 | | 1.00 | |
| Medium | 0.61(0.49–0.75) | 0.001 | 1.61(1.22–2.12) | 0.001 |
| Bad and very bad | 0.57(0.44–0.77) | 0.001 | 1.03(0.74–1.45) | 0.823 |
| **Covid 19 Pandemic effect on family financial status** | | | | |
| Negative and very negative | 1.00 | | 1.00 | |
| Positive and very positive | 1.55(0.84–2.66) | 0.165 | 1.13(0.57–2.25) | 0.880 |
| No effect | 0.72(0.59–0.87) | 0.001 | 0.76(0.58–0.98) | 0.037 |

*(Continued)*

**Table 5.** (Continued)

| Variable | Unadjusted OR(95% CI) | P-Value | Adjusted OR | P-Value |
|---|---|---|---|---|
| **Child working due to the pandemic** | | 0.001 | | 0.032 |
| No | 1.00 | | 1.00 | |
| Yes | 2.16(1.37–3.39) | | 1.93(1.07–3.52) | |
| **Method of education** | | 0.003 | | 0.001 |
| Face-to-face and mix of face-to-face and online | 1.00 | | 1.00 | |
| Online | 1.47 (1.14–1.89) | | 1.40 (1.08–2.20) | |

The findings of our study showed that approximately half of the children wore masks regularly outside the home three months before the study. In Iran, a few studies have measured the use of masks in the general population, mostly involving adults [23–25]. The study by Rahimi, Shirali [23] found that 26.6 percent of pedestrians younger than nine and 45 percent of those between 10 and 39 years old wore masks. Moreover, the study of Ganjali, Soleimani [26] reported that the mask-wearing rate was 61.52% among children aged 2 to 18 years referred to specialized pediatric clinics. In our study, the most common reasons for not wearing masks regularly were the belief that the epidemic was at a low ebb or diminishing (38.9%) and that masks are ineffective in preventing infection (14.6%). Another study reported that 53.50% of the parents of the children who refused to wear masks did not believe in this behavior, and the others had economic problems [26]. Although health authorities of Iran emphasized the necessity of wearing masks as a simple and effective method to avoid COVID-19 infection even after the start of vaccination [27], the requirement was not enforced rigorously.

Regarding wearing a mask, a study in Turkey found that 92% of secondary school students always wore masks outside the house in the spring of 2021, which was a high adherence to mask-wearing. Furthermore, reviewing studies relevant to mask-wearing in children of different countries revealed that child adherence to masking policies was similar in school (range from individual studies: 43%–97%) and community settings (34%–96%), with increased adherence as age increases [28]. Furthermore, due to reviewing studies from Canada, China, Germany, South Korea, and the US, the primary factors linked to low adherence to mask-wearing included: 1) reporting masks were uncomfortable, 2) reporting masks were unattractive, 3) perceived low risk of infection, and 4) negative attitudes toward mask use and its side effects. According to a study of parents with children who attend school in person in grades K-12 at the national level in the US, 46% of parents believe masks have hurt their children's social learning and interactions, about 40% think it has hurt their general schooling experience and mental and emotional health, and about 33% believe it has hurt their education [29]. Interestingly, parents did not mention these concerns about children's mask-wearing in our study. Another study employed the Theory of Planned Behaviour (TPB) to explain factors associated with wearing a mask. TPB focuses on a person's attitudes (i.e., perceptions of pros and cons of mask-wearing), subjective norms (i.e., desire to meet societal norms of mask-wearing), and perceived control (i.e., personal capacity to wear a mask). In this survey of parents of school-aged children in Canada and the US, Moran [30] reported that the intention of parents (with or without children with pre-existing conditions) to get their children to wear masks was impacted by negative attitudes toward mask use, societal norms, and perceived control. Societal norms and intentions predicted mask use in children.

Based on our study results, 54.2 percent of the children in this study received at least one dose of the COVID-19 vaccine, while 36.4 percent received two doses. Since no population-

based study or official report showed vaccination status in Iranian children, we could only compare our findings with the figures the Iranian officials gave in the media. According to Iran's Ministry of Health in June 2022, in the age group of 12–18 years old, more than 75% of the children received at least one dose of the vaccine [30], and according to the Ministry of Education in September 2022, 92% of the children in the same age group received the first dose, and 80% of them received the second dose [31], but it was not reported what percentage of children who were 5–12 years old were vaccinated. It should be noted that the national figures were given for the age group of 12–18 years old, while the present study included the age group of 7–18 years in two cities of Iran.

The most prevalent reason for vaccine hesitancy in our study (44.8%) was fear of vaccine side effects for children. Moreover, while children who were 5–12 years old had been eligible for vaccination since about six months before our study, a third of the cases of vaccine aversion in our study were due to children being in an age group below the threshold for vaccination, as their parents stated. Regression results in our study showed that in the age group of 15–18 years old, non-vaccination was about 80 percent lower than in the age group of 7–14 years old. That is in accordance with studies from Canada [32], Italy [33], and several Arab countries [34] that have also shown a negative association between immunization and being in a younger age group. Overall, a considerable percentage of families did not seem to desire to vaccinate their children, especially younger ones, probably due to a lack of access to reliable and trustworthy information on vaccine benefits and side effects.

A look at the relationship between socio-economic characteristics of families and child vaccination against COVID-19 in our study shows that the more educated the mothers were, the less likely they were to resist child vaccination. Moreover, families with good financial situations were 72 percent less likely to resist child vaccination, while the odds ratio for low-income families to oppose child vaccination against COVID-19 was 1.4. Studies in other countries also have similar findings. A nationwide survey in the US [35] and the UK [36] before vaccination showed that parents who resisted vaccination were more concerned with the vaccine's safety than its effectiveness. In contrast, lower-income families were more resistant than others. In Canada, the leading causes of parental resistance to COVID-19 vaccination for children were concerns about its safety and side effects and lower-income families being more reluctant to vaccinate their children [32].

In investigating the role of being an immigrant in child vaccination, the odds ratio of non-vaccination of Afghan children was 2.77 compared to Iranian children. Our findings also showed that 24.6% of Afghan participants, compared to 4.7% of Iranian respondents, believed vaccination was ineffective against COVID-19. In this regard, a study in Canada found higher rates of non-vaccination in children 12–17 years old in immigrant communities compared to non-immigrants [37]. Other studies have highlighted factors associated with COVID-19 vaccination in immigrants, including the inequality in vaccination and lower access to health care in low-income groups, such as refugees and immigrants, and barriers such as lack of knowledge of the benefits and risks of vaccines, social norms (such as the concept of "fatalism" expresses the belief that everything that happens in life is predetermined, it is not possible to go beyond this predetermination) [38] and effects, lack of information on how to access to the vaccine, and fear of being detained, imprisoned or deported [39].

The odds ratio of non-vaccinated children living in Karaj was approximately 3.5 compared to Tehran's. In our study, Afghan children accounted for 8% of the total sample, and the proportion of Afghan children in the Karaj sample was twice compared to that of Tehran (13% vs.

6%). Therefore, a higher rate of non-vaccination in Karaj could be due to the higher percentage of Afghan children, at least partially.

Our findings showed that three months before the interview, approximately 20 percent of children had been infected with COVID-19, of whom 1.4 percent were hospitalized. Hence, COVID-19 was reported in more than half of our study sample according to only clinical symptoms. In Iran, there are no national statistics available on children infected with COVID-19. A study of 15,000 urban and rural children and adults across Iran estimated that from the pandemic's start to December 2020, 9.6% of cases were children under 18 years old, including those who died, which was 2.5% of the total population of children [22]. Then, the higher prevalence of COVID-19 in our study could be due to the infection report being based only on clinical symptoms. It should be considered that in Iran when a family member was infected with COVID-19, other members with similar symptoms did not take the test due to the relatively high cost of PCR testing. In addition to the method of diagnosis, our study coincided with school reopening and the seventh wave of COVID-19 in Iran.

The regression model of the study showed that COVID-19 infection was significantly higher in the age group of 15–18 (28%) compared to the age group of 7–14 (17%). Still, there was no significant difference between girls and boys, two cities of Tehran and Karaj, and Iranian and Afghan children. Along with our study, global findings also showed that younger children are less likely to contract COVID-19: Out of 75 million under 20 cases infected with COVID-19 worldwide, 39 percent involved children younger than nine [3]. Moreover, WHO [40] reported that during the initial pandemic phase with the ancestral strain, from December 2019 to October 2021, children under five years of age represented 2% of reported global cases and 0.1% of reported global deaths. According to a recent review study, key mechanisms that protect young children from severe COVID-19 disease include the placental barrier, differential expression of angiotensin-converting enzyme 2 (ACE-2) receptors, immature immune response, and passive transfer of antibodies via placenta and human milk [41]. Older children and younger adolescents (5 to 14 years) account for 7% of reported global cases and 0.1% of reported global deaths. Additionally, it is reported that COVID-19 cases among children spiked dramatically in 2022 during the Omicron variant surge at a time when most countries relaxed public health and social measures. Globally, by July 2022, children below the age of 5 years and those aged 5–14 years presented 2.47% and 10.44%, respectively. Adolescents and young adults aged 15 to 24 presented 13.91% of all cases [40].

In investigating the relationship between the socioeconomic characteristics of families and the proportion of children infected with COVID-19, we found that permanent unemployment, unstable employment, and retirement of family heads increased the chance of children contracting COVID-19 as much as 2.6, 1.7, and 2.1, respectively. The odds ratio of COVID-19 infection in children with a moderate financial status was 1.6 compared with families with a good economic status. Since the outbreak of the COVID-19 pandemic, there have been reports about the possibility of the socio-economic impact of COVID-19 on children. For example, the proportion of cases of infection among those under 20 years old was about 11 percent in low- and middle-income countries, compared to 7 percent in high-income countries [6]. The higher prevalence of COVID-19 in children of low-income countries and low-income families is related to malnutrition, lower access to health services, and living in densely populated environments, which are exacerbated due to family unemployment, school closures, and health services suspension as a result of the COVID-19 outbreak [6, 42]. Besides, some evidence globally has shown that the pandemic has increased financial inequalities between and within countries [2, 43–45]. In general, low-income families during the pandemic have experienced decreased access to appropriate services in all aspects of health care and social support services

due to lockdowns and social distancing policies, which put their children at more risk than others [46].

In Iran, some studies and reports have predicted a worsening of financial conditions following the onset of the pandemic, especially for lower-income families, and reported that the impact of the pandemic would be more severe for this group because of lower income, seasonal, and part-time employment, and lower social security coverage [47]. Furthermore, a qualitative study on low-income families during the coronavirus pandemic also showed that the pandemic greatly exacerbated the adverse situation of children's education, health, and nutrition [48]. According to our findings, the pandemic has increased the chances of children contracting COVID-19 by worsening families' financial situation (in half of our sample) and making it challenging to meet the needs of children (44%). The ability of 43 percent of families to pay for the health costs of children worsened as a result of COVID-19, so that 10 percent of 22.5 percent of children who needed health care services, including COVID-19, in three months before the study, did not receive any treatment because of the high cost of treatment. In addition, nearly a quarter of children were not covered by health insurance, while just a tiny percentage were covered by the services of support organizations.

## Conclusions

The COVID-19 pandemic in Iran, as in many other countries worldwide, has contributed to unemployment and financial hardship for low-income families. Declining family income and increasing difficulty in meeting various needs have made their children more vulnerable to health problems, a lack of social and insurance support, and lower access to health services, including those related to COVID-19, which have increased COVID-19 infection. In addition, a considerable percentage of low-income families and immigrants were reluctant to vaccinate their children, resulting in a higher COVID-19 infection among these groups. The government's failure to adequately meet the financial needs of the people during the pandemic made it less strict about implementing preventive measures, such as the closure of jobs and using masks. Because of this, many businesses are allowed to be open without any fines, leading to increased COVID-19 infection among children and adults. The continuing financial problems of a significant portion of families and the involvement of more children from low-income families with the disease in the third year of the pandemic indicate that the decline of the pandemic is not the end of its problems.

## Author Contributions

**Conceptualization:** Meroe Vameghi, Giti Bahrami, Farin Soleimani, Marzieh Takaffoli.

**Data curation:** Meroe Vameghi, Mohammad Saatchi, Giti Bahrami.

**Formal analysis:** Meroe Vameghi, Mohammad Saatchi.

**Funding acquisition:** Meroe Vameghi, Giti Bahrami.

**Investigation:** Giti Bahrami.

**Methodology:** Meroe Vameghi, Mohammad Saatchi.

**Project administration:** Mohammad Saatchi, Marzieh Takaffoli.

**Software:** Mohammad Saatchi.

**Supervision:** Meroe Vameghi, Mohammad Saatchi, Farin Soleimani.

**Validation:** Meroe Vameghi, Mohammad Saatchi.

**Writing – original draft:** Mohammad Saatchi, Giti Bahrami, Marzieh Takaffoli.

**Writing – review & editing:** Giti Bahrami, Farin Soleimani, Marzieh Takaffoli.

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
