## [Decision Letter · Decision Letter 0]

28 Feb 2024

PONE-D-23-42991How did we protect children against Covid-19 in Iran?  Prevalence of COVID-19 and vaccination in the socio-economic context of an epidemicPLOS ONE

Dear Dr. Saatchi, 

Thank you for submitting your manuscript to PLOS ONE. After careful consideration, we feel that it has merit but does not fully meet PLOS ONE’s publication criteria as it currently stands. Therefore, we invite you to submit a revised version of the manuscript that addresses the points raised during the review process.

Please submit your revised manuscript by Apr 13 2024 11:59PM. If you will need more time than this to complete your revisions, please reply to this message or contact the journal office at plosone@plos.org. Please include the following items when submitting your revised manuscript:A rebuttal letter that responds to each point raised by the academic editor and reviewer(s). You should upload this letter as a separate file labeled 'Response to Reviewers'.A marked-up copy of your manuscript that highlights changes made to the original version. You should upload this as a separate file labeled 'Revised Manuscript with Track Changes'.An unmarked version of your revised paper without tracked changes. You should upload this as a separate file labeled 'Manuscript'.We look forward to receiving your revised manuscript.

Kind regards,

Nour Amin Elsahoryi, pHD

Academic Editor

PLOS ONE

Journal Requirements:

Whilst you may use any professional scientific editing service of your choice, PLOS has partnered with both American Journal Experts (AJE) and Editage to provide discounted services to PLOS authors. Both organizations have experience helping authors meet PLOS guidelines and can provide language editing, translation, manuscript formatting, and figure formatting to ensure your manuscript meets our submission guidelines. To take advantage of our partnership with AJE, visit the AJE website (http://aje.com/go/plos) for a 15% discount off AJE services. To take advantage of our partnership with Editage, visit the Editage website (www.editage.com) and enter referral code PLOSEDIT for a 15% discount off Editage services. If the PLOS editorial team finds any language issues in text that either AJE or Editage has edited, the service provider will re-edit the text for free.

This study was sponsored by the University of Social Welfare and Rehabilitation Science  [contract number: 1401/801/A/11799 ] (https://uswr.ac.ir/) and Alborz University of Medical Sciences, Karaj, Iran [contract number: 1401/60/D/1495] (https://abzums.ac.ir/).

Reviewers' comments:

Reviewer's Responses to Questions

**Comments to the Author**

1. Is the manuscript technically sound, and do the data support the conclusions?

Reviewer #1: Yes

Reviewer #2: Yes

2. Has the statistical analysis been performed appropriately and rigorously? 

Reviewer #1: Yes

Reviewer #2: Yes

3. Have the authors made all data underlying the findings in their manuscript fully available?

Reviewer #1: Yes

Reviewer #2: Yes

4. Is the manuscript presented in an intelligible fashion and written in standard English?

Reviewer #1: No

Reviewer #2: Yes

5. Review Comments to the Author

Reviewer #1: This paper is the kind that provides substantive evidence for things we assume are true. As such, although it is not groundbreaking, such work is essential to the work of others, such as those in the scientific community and those in the government and non-governmental organizations who can effect changes that address this paper's conclusions. On this basis, I believe the work should be published.

However, it needs a thorough copy editing. I found the following errors, some of which are clear errors, while others are simply ambiguities. Errors must be corrected before the paper is published. Raw data is not provided, so it is not possible to check the data, but I strongly urge that the authors recheck key statistics. A correction or retraction would be detrimental to the authors themselves, as well as calling into question the conclusions in the paper.

Specific issues I found:

66 >> Some references related to evidence of support for the paper's assumptions were to opinion pieces, not to research articles. (9 and 10 for example). References to support statements of fact should be to research that contains that evidence.

76 >> "Feb 2019" as start of epidemic is wrong

232 The report of definitely unwillingness among Iranians living in Tehran and Karaj was 65% and 29%, respectively, and among Afghans, it was 41% and 24%

>> "definite unwillingness" should be clarified. The figure is missing important axis and legend information.

312 >> it's--its

418 In Iran, some studies and reports have predicted an escalation of financial conditions following the onset of the pandemic

>> "escalation" does not make sense in this context. Escalation would be an increase. It should be a "worsening" or an "escalation in poverty".

440 The government's failure to adequately meet the financial needs of the people during the pandemic made it less strict about implementing preventive measures, such as the closure of jobs and using masks, which led to an increase in COVID-19 infection among children and adults.

>> No support for how a failure to meet financial needs made it less strict about implementing preventive measures. Meaning unclear.

Reviewer #2: 1. The manuscript was technically sound and the data supported its conclusion.

2. The statistical analysis has been performed appropriately and rigorously.

3. The authors have made all data to be fully available.

4. The manuscript was presented in an intelligible fashion and written in standard English.

6. PLOS authors have the option to publish the peer review history of their article (what does this mean?). If published, this will include your full peer review and any attached files.

Reviewer #1: **Yes: **Aaron H. Abend

Reviewer #2: **Yes: **Kassa Demissie (PhD)

---

## [Author Response · Author response to Decision Letter 0]

14 Apr 2024

A- Response to the Editor

Comment

Response: The final version of the manuscript was checked and submitted according to the instructions.

 Comment

"We suggest you thoroughly copyedit your manuscript for language usage, spelling, and grammar. If you do not know anyone who can help you do this, you may wish to consider employing a professional, scientific editing service" 

Response: Javad Ghholami, Virayeshyar English language editing and Translation center

Comment

Please provide an amended statement that declares all the funding or sources of support

Response: This study sponsored by the University of Social Welfare and Rehabilitation Science [contract number: 1401/801/A/11799 ] (https://uswr.ac.ir/) received by Meroe Vameghi and Alborz University of Medical Sciences, Karaj, Iran [contract number: 1401/60/D/1495] (https://abzums.ac.ir/) received by Giti Bahrami. Moreover, no additional external funding was received for this study.

Comment

PLOS requires an ORCID iD for the corresponding author in Editorial Manager on papers submitted after December 6th, 2016.

Response: ORCID id of Dr. Mohammad Saatchi: 0000-0003-2744-9927

Unfortunately, we cannot log in to ORCID account on the first page of the Editorial Manager. We face this error:

B- Response to the Reviewers' comments:

Reviewer #1:

However, it needs a thorough copy editing. I found the following errors, some of which are clear errors, while others are simply ambiguities. Errors must be corrected before the paper is published. Raw data is not provided, so it is not possible to check the data, but I strongly urge that the authors recheck key statistics. A correction or retraction would be detrimental to the authors themselves, as well as calling into question the conclusions in the paper.

Author: We checked all data again. 

66>> Some references related to evidence of support for the paper's assumptions were to opinion pieces, not to research articles. (9 and 10 for example). References to support statements of fact should be to research that contains that evidence.

Author: Using Opinions of government officials instead of research findings is due to a lack of governmental official reports and any research results on many aspects of the COVID-19 pandemic in Iran, including the prevalence of COVID-19 infection, vaccination, and vaccine hesitancy among children.

74>>"Feb 2019" as start of epidemic is wrong

Author: Feb 2019 was corrected to Feb 2020

232>>The report of definitely unwillingness among Iranians living in Tehran and Karaj was 65% and 29%, respectively, and among Afghans, it was 41% and 24%

>> "definite unwillingness" should be clarified. The figure is missing important axis and legend information.

Author: There is no definition for " definite unwillingness ", but as an option of a person's desire to receive a vaccine in the future, it indicates that people, according to their feelings, understanding, and existing conditions, not only have not received a vaccine so far, but in the future, they definitely do not want to receive the Covid-19 vaccine.

307>>it's—its

Author: It was corrected to its

415>> In Iran, some studies and reports have predicted an escalation of financial conditions following the onset of the pandemic. “escalation" does not make sense in this context. Escalation would be an increase. It should be a "worsening" or an "escalation in poverty".

Author: “Escalation” was corrected to “worsening”

437>> The government's failure to adequately meet the financial needs of the people during the pandemic made it less strict about implementing preventive measures, such as the closure of jobs and using masks, which led to an increase in COVID-19 infection among children and adults. >> No support for how a failure to meet financial needs made it less strict about implementing preventive measures. Meaning unclear.

Author: This means that because the government could not support businesses' closure during the pandemic, it allowed businesses to remain open despite the high prevalence of COVID-19. This situation led to an increase in COVID-19 infection among children and adults. We corrected this part with clearer phrases.

Reviewer#2

1>> This should be a research question and should be placed in the methodology section.

Author: We add this to method section.

2>> COVID-19 epidemic in Iran

Author: We changed the title based on the comment

27>> Mention its version number

Author: We add the version number

104> Did the study cover all districts in both sites?

Author: Yes. As we explained, 22 districts of Tehran and 12 districts of Karaj were investigated using cluster sampling.

106>> What was your unit of sampling? It is a household.

Author: Exactly. In each cluster, the household was the study unit. It added. 

134>> House ownership 

Author: It changed to house ownership

176> Your observation was 2878. But it is not for instance in the case of child sex, etc

Why?

Author: The missing data for each of the variables in the study makes the number of variables not equal to the total number.

181>>Income = 98.3% Child’s needs = 98% Why not 100%? 

Author: We couldn’t understand the comment

190>> Table 2

Author: All comments for Table 2 regarding the difference between the numbers of the total sample of study and the numbers in the table are because of missing data

213>> 61% + 41% = 102%, why?

Author: These two values should not be added together. One is the percentage for Tehran, and the other is the percentage for Karaj. In total, 54% of participants have received at least one dose of vaccine.

214>> 42% + 83.2% = 125.2%?

Author: These two values should not be added together. One is the percentage of vaccination coverage in the 7-14 age group, and the other is the percentage of vaccination coverage in 15-18 age group

231>>1206 in table 3.

Author: It is corrected 

279>> To say representative, the target and study populations would have been indicated. You only indicated the general sampled study population. The target population was not indicated. Since it was randomly sampled, I think it is representative.

Author: The target population in the present study was families with children under 18 years of age in Tehran. Using random sampling, we tried to select a representative sample of these households.

335>> What is the immunological reason for this?

Author: We don’t consider the causes of this difference according to immunological reasons, but rather deciding to no vaccinating children is the result of harm estimation for infection by covid-19 and vaccination out comes for two age groups.

345>> Is it to mean similar findings?

Authors: Yes, it is, and it is corrected. 

359>> How?

Author: According to some societal norms, vaccination could not change the predetermined death for people. We changed the sentence to: “Social norms (such as the concept of “fatalism” expresses the belief that everything that happens in life is predetermined, it is not possible to go beyond this predetermination) (Erkal et al, 2023)”

388>> Why? What is the Immunological mechanism involved?

Author: We added the immunologic reasons to the paper, but it seems somehow unrelated to a discussion based on a more sociologic interpretation of findings, and we prefer not to add this part to the paper. 

“According to a recent review study, key mechanisms that protect young children from severe COVID-19 disease include the placental barrier, differential expression of angiotensin-converting enzyme 2 (ACE-2) receptors, immature immune response, and passive transfer of antibodies via placenta and human milk.”

---

## [Editor Report · Decision Letter 1]

17 Apr 2024

How did we protect children against COVID-19 in Iran?

Prevalence of COVID-19 and vaccination in the socio-economic context of COVID-19 epidemic

PONE-D-23-42991R1

Dear Dr. Saatchi,

We’re pleased to inform you that your manuscript has been judged scientifically suitable for publication and will be formally accepted for publication once it meets all outstanding technical requirements.

Kind regards,

Nour Amin Elsahoryi, pHD

Academic Editor

PLOS ONE

Additional Editor Comments (optional):

Thank you for your cooperation.

I have thoroughly reviewed all comments and the author's response, and I believe they have addressed all the necessary points and edited the manuscript accordingly.

Regards,

Dr. Nour Elsahoryi

---

## [Editor Report · Acceptance letter]

29 Apr 2024

PONE-D-23-42991R1 

PLOS ONE

Dear Dr. Saatchi, 

I'm pleased to inform you that your manuscript has been deemed suitable for publication in PLOS ONE. Congratulations! Your manuscript is now being handed over to our production team.

Kind regards, 

on behalf of

Dr. Nour Amin Elsahoryi 

Academic Editor

PLOS ONE